# Post-Cholecystectomy Right Hepatic Artery Pseudoaneurysm Induced by Generalized Seizure: A Case Report

**DOI:** 10.3390/medicina58050568

**Published:** 2022-04-21

**Authors:** Yueh-Hsun Tsai, Hao-Ming Chang

**Affiliations:** 1Department of Surgery, Tri-Service General Hospital, National Defense Medical Center, Taipei 11490, Taiwan; drxelamai@gmail.com; 2Division of General Surgery, Department of Surgery, Tri-Service General Hospital, National Defense Medical Center, Taipei 11490, Taiwan

**Keywords:** laparoscopic cholecystectomy, pseudoaneurysm, seizure, transcatheter arterial embolization, right hepatic artery pseudoaneurysm, case report

## Abstract

Pseudoaneurysm is a rare complication of laparoscopic cholecystectomy (LC). In most cases, the patient presents with gastrointestinal bleeding or hemoperitoneum. Here, we present a case with a post-cholecystectomy right hepatic artery pseudoaneurysm (PSA) induced by a generalized seizure. A 39-year-old male was sent to the emergency room with a generalized seizure and a loss of consciousness for approximately 5 min. Diffuse abdominal pain was complained of after consciousness returned. The surgical history of LC 13 days prior was mentioned. Abdominal computer tomography (CT) revealed a lobulated fluid accumulation in the gallbladder fossa with prominent fatty stranding and suspected biloma formation. After admission for one week, sharp abdominal pain was observed. Abdominal CT angiography revealed a right hepatic artery pseudoaneurysm. Transcatheter arterial embolization was performed with a total of seven platinum coils. In conclusion, it is important for doctors to take pseudoaneurysm into consideration in the patient who presents with seizure attack after receiving LC. Late discovery of PSA when it is ruptured can lead to fatal conditions, such as severe hemoperitoneum.

## 1. Introduction

Laparoscopic cholecystectomy (LC) is the standard surgery for symptomatic cholelithiasis. Injury to the vascular wall can occur during the procedure. According to previous studies, the incidence of bile duct injuries and bowel injuries during LC were reported to be approximately 0.6% [1] and 0.2% [2], respectively. Sometimes delayed onset of artery bleeding can lead to right hepatic artery or cystic artery pseudoaneurysm (PSA) formation. Patients who have postoperative PSA may develop hemobilia or hemoperitoneum. Here, we present a case with a PSA of the right hepatic artery after an LC associated with a seizure attack.

## 2. Detailed Case Description

A 39-year-old male without systemic disease was sent to the emergency room with generalized seizure and conscious loss for approximately 5 min. It was his first episode of seizure attack, according to his family. After recovery of consciousness, diffuse abdominal pain was observed. The surgical history of LC 13 days ago was mentioned. The decision to perform laparoscopic cholecystectomy was under the diagnosis of gallstones with chronic cholecystitis to wall off a perforation and abscess formation. The preoperative abdominal CT revealed a gallbladder with a stone (size: 4.5 cm) with wall thickening and peri-cholecystic fatty stranding. No abnormal findings in the liver, biliary system, pancreas, and spleen were noted. The laparoscopic cholecystectomy was performed smoothly, and the patient tolerated the whole process well. After the surgery, a fever episode was noted on the first postoperative day, and atelectasis was suspected because the patient did not have abdominal pain or abdominal tenderness. Under empiric antibiotics treatment, the fever subsided, and the patient was discharged on the fourth postoperative day. At the emergency department, the patient presented with hypotension and sinus tachycardia, which responded to fluid resuscitation. Physical examination revealed tenderness of the whole abdomen with muscle guarding. Blood examinations showed leukocytosis (24,330/dL), hemoglobin level of 9.3 g/dL (dropped from 13.7 g/dL in two weeks), normal total bilirubin (0.4 mg/dL), elevated C-reactive protein (2.17 mg/dL), and lactate level (3.3 mmol/L). No intracranial hemorrhage was discovered from computer tomography (CT) of the brain. Abdominal CT revealed lobulated fluid accumulation in the gallbladder fossa with prominent fatty stranding and suspected biloma formation induced by a generalized seizure. (Figure 1) The patient was admitted under the impression of a generalized seizure with intra-abdominal infection. After admission, electroencephalography (EEG) revealed normal results. Empiric antibiotics with Tapimycin 4.5 g (Piperacillin 4 g + Tazobactam 500 mg) Q6H were prescribed. On the second week of admission, worsening abdominal pain with intermittent fever was noted. Abdomen CT angiography revealed a 3 cm × 2.7 cm × 2.7 cm PSA formation over the right hepatic artery with hemoperitoneum (Figure 2). Transcatheter arterial embolization (TAE) was performed on the inferior branch of the right hepatic artery with a total of seven platinum coils (Figure 3). The percutaneous drainage of the intra-abdominal hematoma was arranged again after TAE. The patient improved gradually with adequate drainage and antibiotic treatment. The patient was discharged three weeks after TAE.

## 3. Discussion

A review of 14,243 patients who underwent different standard laparoscopic procedures showed an overall hemorrhage rate of 4.1%, with bleeding rates of 2.3% intraoperatively and 1.8% postoperatively [3]. Leakage from an injured artery into the surrounding tissues causes a PSA outside the artery [4]. The most common vessels involved in post-LC PSA are the right hepatic artery, which accounts for 87.1% of cases; the cystic artery, reported in 7.9% of cases; both the cystic and hepatic arteries in 4.0% of cases; and the gastroduodenal artery (1.0%) [5].

The pathophysiology of right hepatic/cystic artery aneurysm varies from vascular laceration during the surgery, thermal injury, or clip application. Mechanical injury can easily occur when dissecting Calot’s triangle, especially if the adhesion is severe. If an inadvertent thermal injury is caused by cautery or indirectly via a clip during the procedure, it may cause scarring of the vessel, causing a PSA after detachment [6,7]. The diagnosis of PSA is often made when the patient is symptomatic. Patients may present with upper gastrointestinal (GI) bleeding or hemoperitoneum when the PSA is large. The classic presentation of hemobilia is Quincke’s triad (jaundice, abdominal pain, and GI bleeding), but it is only observed in 25–30% of cases [8]. In our case, the patient initially presented with a seizure attack and then developed IAI with sepsis. Biloma formation was found from the initial abdominal CT. The patient was initially diagnosed with a seizure attack and IAI. PSA was discovered when the patient developed hemoperitoneum and severe sepsis. A possible explanation is that the PSA may have developed following the detachment of the vessel scar during the seizure attack. Bleeding from the PSA accumulated in the gallbladder fossa caused an IAI, but the PSA was too small to be detected by CT. TAE is the first-line treatment for PSA when feasible. The success rate of TAE was reported to be 80–100%, with lower mortality and morbidity rates than surgery [9].

## 4. Conclusions

In conclusion, PSA is a rare condition after LC. It is important for doctors to take pseudoaneurysm into consideration in patients who present with seizure attack after receiving LC. Late discovery of PSA when it is ruptured can lead to fatal conditions. TAE can be performed securely in most early cases.

## Figures and Tables

**Figure 1 medicina-58-00568-f001:**
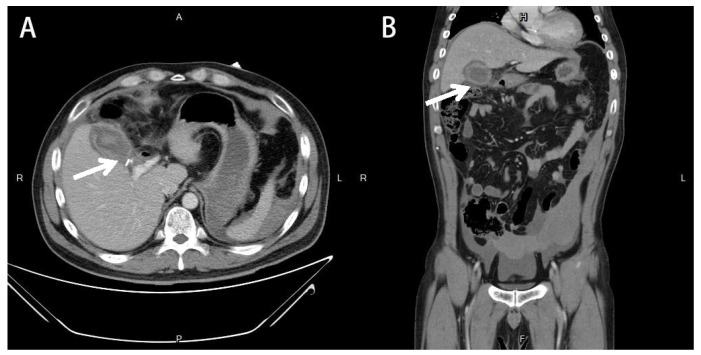
Abdominal computed tomography (CT) images two weeks after laparoscopic cholecystectomy. (**A**) Axial view showing lobulated fluid accumulation in the gallbladder fossa with prominent fatty stranding with clip retention after laparoscopic cholecystectomy (arrow). (**B**) Coronal view showing a lobulated fluid accumulation in the gallbladder fossa with prominent fatty (arrow).

**Figure 2 medicina-58-00568-f002:**
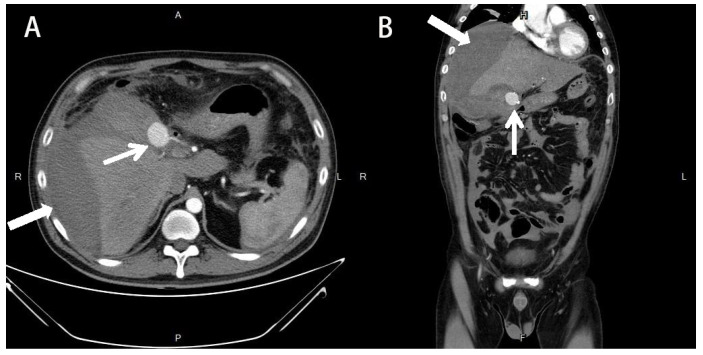
Abdominal arterial-phase CT scan taken four weeks after laparoscopic cholecystectomy. (**A**) Axial view showing a 3 cm × 2.7 cm pseudoaneurysm formation over the right hepatic artery (thin arrow) with hemoperitoneum (thick arrow). (**B**) Coronal view showing a 2.7 cm × 2.7 cm pseudoaneurysm formation over the right hepatic artery (thin arrow) with hemoperitoneum (thick arrow).

**Figure 3 medicina-58-00568-f003:**
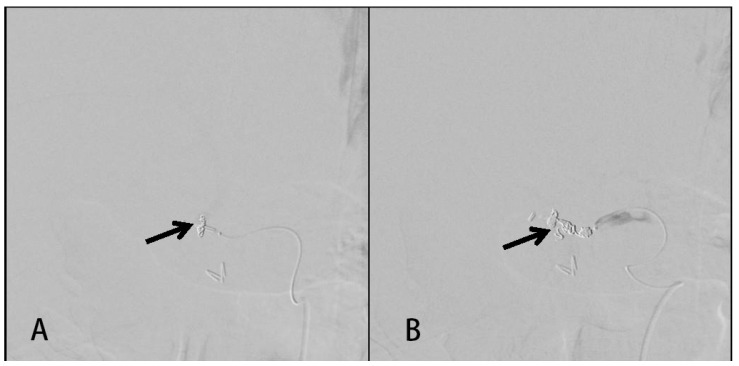
Transcatheter arterial embolization of the right hepatic artery. (**A**) Transcatheter arterial (TAE) showed a contrast-filling outpouching (arrow) at the site of the inferior branch of the right hepatic artery near the surgical clips of cholecystectomy. (**B**) TAE was performed with a total of seven platinum coils (arrow).

## Data Availability

Not applicable.

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
