# Peer review of "Post-Cholecystectomy Right Hepatic Artery Pseudoaneurysm Induced by Generalized Seizure: A Case Report"

_medicina, 2022, doi:10.3390/medicina58050568_

Round 1
Reviewer 1 Report
The presented case is dedicated to rare complication of cholecystectomy. However, some more data and corrections are required.
- The patient was presented with seizures. The head ct excluded the hemorrhage. The authors concluded that the seizures were attributed to infection and sepsis. C-reactive protein was mildly elevated and there is no data about procalcitonin. Diagnosis of IAI is questionable
- What was the indication for LC.
- More data about primary surgery is required (eg uncomplicated course with discharge in 24 hours etc)
After those correction the paper could be recommended for the Journal
Author Response
- The patient was presented with seizures. The head ct excluded the hemorrhage. The authors concluded that the seizures were attributed to infection and sepsis. C-reactive protein was mildly elevated and there is no data about procalcitonin. Diagnosis of IAI is questionable.
Response:
Dear reviewer, thanks for your comment. In our study, procalcitonin was not tested because infection was not the impression at first when the patient was sent to the hospital presenting with generalized seizure. However, abdomen CT revealed a lobulated fluid accumulation in the gallbladder fossa with prominent fatty stranding, suspected biloma formation after seizure attack. The patient also presented with fever (>38.0°C), hypotension, and abdominal. Hence, diagnosis of intra-abdominal infection (IAI) was made. However, in this case, with lab data of procalcitonin, diagnosis of IAI could be more accurate.
- What was the indication for LC.
Response:
Dear reviewer, thanks for your comment. The decision to perform laparoscopic cholecystectomy was under diagnosis of gallstones with chronic cholecystitis, with wall off perforation and abscess formation.
- More data about primary surgery is required (eg uncomplicated course with discharge in 24 hours etc)
Response:
Dear reviewer, thanks for your comment. The laparoscopic cholecystectomy was performed smoothly and the patient tolerated the whole process well. After the surgery, a fever episode was noted on postoperative day 1, and atelectasis was suspected because the patient didn't have abdominal pain nor abdominal tenderness. Under empiric antibiotics treatment, fever subsided and the patient discharged on postoperative day 4.
Reviewer 2 Report
I would like to suggest the authors to sligthly modify the title, because it seems that the modality of PSA appearance was the seizure.
Thank you very much for this interesting case of unusual but ominous complication of LC.
Please modify line 45 putting a space between "was" and "suspected"
Author Response
- I would like to suggest the authors to slightly modify the title, because it seems that the modality of PSA appearance was the seizure. Thank you very much for this interesting case of unusual but ominous complication of LC. Please modify line 45 putting a space between "was" and "suspected".
Response:
Dear reviewer, thanks for your comment. We have made the correction in our manuscript. We also changed the title to “Post-cholecystectomy right hepatic artery pseudoaneurysm presents with induced by generalized seizure: a case report” according to your suggestion.
Reviewer 3 Report
In the present case report Tsai et al described a patient who, 13 days after laparoscopic cholecistectomy (LC), developed a pseudo-aneurysm of hepatic artery with rupture and hemoperitoneum, which presented at first with seizures. Main comments:
1) I do not feel that this case report has a particular interest, since it simply report a delayed complication of LC and does not bring any novelty, nor useful message for the readers.
2) How many days was percutaneous drainage kept in situ?
3) Which empiric antibiotics were prescribed?
4) Considering the recent history of LC, why abdominal CT scan was performed only one week after admission?
Author Response
- I do not feel that this case report has a particular interest, since it simply report a delayed complication of LC and does not bring any novelty, nor useful message for the readers.
Response:
Dear reviewer, thanks for your comment. In our case, the patient initially presented with seizure attack then developed IAI with sepsis. The symptoms were not the typical manifestations of hepatic artery pseudoaneurysm. Besides, pseudoaneurysm was not found from the initial abdominal CT. A possible explanation is that the PSA may have developed following the detachment of the vessel scar at admission. We think this is an interesting case with uncommon clinical course. Also it is important for doctors to take pseudoaneurysm into consideration in patient who presents with seizure attack after receiving LC even if the patient present with atypical symptoms.
- How many days was percutaneous drainage kept in situ?
Response:
Dear reviewer, thanks for your comment. The percutaneous drainage was kept for 3 weeks. We removed it after the fever subsided with very few daily drainage amount.
- Which empiric antibiotics were prescribed?
Response:
The patient was treated with Tapimycin 4.5g (Piperacillin 4g + Tazobactam 500mg) Q6H during admission. However, we switched to Tigecycline 50 mg Q12H for 2 courses due to persistent fever.
- Considering the recent history of LC, why abdominal CT scan was performed only one week after admission?
Response:
Dear reviewer, thanks for your comment. In this case, intermittent fever was noted after admission. Worsened abdominal pain was noted on the second week after admission. Based on his surgical history and unstable vital signs, new developed vessel injury was suspected. Also, physical examinations revealed signs of peritonitis such as abdominal rigidity and rebound tenderness. Hence, abdominal CT was performed again for evaluation of internal bleeding.
Round 2
Reviewer 1 Report
The authors improved the paper and it can be reccomened for publications with minor revisions.
Does the patient had epilepsy or it was the first episode in his life? Have to be clarified.
Line 43 have to be corrected (English)
Author Response
- The authors improved the paper and it can be recommended for publications with minor revisions. Does the patient had epilepsy or it was the first episode in his life? Have to be clarified. Line 43 have to be corrected (English)
Response: Dear reviewer, thanks for your comment.
- Yes, it was the patient’s first episode of seizure attack in his life. We have clarified in the revised manuscript.
- Thanks for reminding. We have corrected the sentence into “The decision to perform laparoscopic cholecystectomy was under diagnosis of gall-stones with chronic cholecystitis, wall off perforation and abscess formation.”

Reviewer 3 Report
no further comments
Author Response
Dear reviewer, thanks for taking your time reviewing our study. We have made minor correction according to other reviewer in this version of manuscript. Thank you!